# A Layered View on Focal Adhesions

**DOI:** 10.3390/biology10111189

**Published:** 2021-11-16

**Authors:** Karin Legerstee, Adriaan B. Houtsmuller

**Affiliations:** Department of Pathology, Erasmus MC, Doctor Molewaterplein 40, 3015 GD Rotterdam, The Netherlands; klegerstee@gmail.com

**Keywords:** cell adhesion, cell migration, stratified structure, nanoscale, protein dynamics, super resolution microscopy

## Abstract

**Simple Summary:**

The cytoskeleton is a network of protein fibres within cells that provide structure and support intracellular transport. Focal adhesions are protein complexes associated with the outer cell membrane that are found at the ends of specialised actin fibres of this cytoskeleton. They mediate cell adhesion by connecting the cytoskeleton to the extracellular matrix, a protein and sugar network that surrounds cells in tissues. Focal adhesions also translate forces on actin fibres into forces contributing to cell migration. Cell adhesion and migration are crucial to diverse biological processes such as embryonic development, proper functioning of the immune system or the metastasis of cancer cells. Advances in fluorescence microscopy and data analysis methods provided a more detailed understanding of the dynamic ways in which proteins bind and dissociate from focal adhesions and how they are organised within these protein complexes. In this review, we provide an overview of the advances in the current scientific understanding of focal adhesions and summarize relevant imaging techniques. One of the key insights is that focal adhesion proteins are organised into three layers parallel to the cell membrane. We discuss the relevance of this layered nature for the functioning of focal adhesion.

**Abstract:**

The cytoskeleton provides structure to cells and supports intracellular transport. Actin fibres are crucial to both functions. Focal Adhesions (FAs) are large macromolecular multiprotein assemblies at the ends of specialised actin fibres linking these to the extracellular matrix. FAs translate forces on actin fibres into forces contributing to cell migration. This review will discuss recent insights into FA protein dynamics and their organisation within FAs, made possible by advances in fluorescence imaging techniques and data analysis methods. Over the last decade, evidence has accumulated that FAs are composed of three layers parallel to the plasma membrane. We focus on some of the most frequently investigated proteins, two from each layer, paxillin and FAK (bottom, integrin signalling layer), vinculin and talin (middle, force transduction layer) and zyxin and VASP (top, actin regulatory layer). Finally, we discuss the potential impact of this layered nature on different aspects of FA behaviour.

## 1. Introduction

### 1.1. Focal Adhesions and the Actin Cytoskeleton

The driving force required for the motility of eukaryotic cells is primarily produced by actin polymerisation and the force on actin filaments generated by connected contractile myosin [1]. Translating the force on actin fibres into cell movement is mainly taken care of by Focal Adhesions (FAs), macromolecular multiprotein assemblies at the end of actin fibres that connect them to the extracellular matrix (ECM) [2] (Figure 1A). The actin fibres linked to FAs are known as stress-fibres, a specialised form of F-actin or filamentous actin associated with contractile myosin II filaments [2] (Figure 1B). They are typically composed of approximately 10 to 30 actin filaments primarily crosslinked by α-actinin [3,4]. Other actin-crosslinking proteins associated with stress fibres include filamin and fascin [5,6]. Force is produced by the ATP-driven movement of the myosin filaments along the polarised actin filaments, which results in contraction of the stress fibre and a pulling force on the FA complex [7]. Traditionally, two types of stress fibres associated with FAs are recognised. Ventral stress fibres are associated with FAs at either end and typically transverse through the whole cell [8,9]. Dorsal stress fibres are linked to FAs on one end, typically near the cell front, then stretch upwards to the nucleus and the dorsal cell surface [8]. Recently, a third type of stress fibres associated with FAs was recognised, the cortical stress fibres, which are connected to FAs at both ends but are thinner and less contractile than ventral stress fibres [7].

The cytoskeleton provides cells with structure and supports intracellular transport; actin filaments are crucial to both functions. The cytoskeleton is directly linked to the ECM by FAs, which are, therefore, continuously exposed to force. The force experienced by FAs depends on the combination of myosin-II contractility, which determines the pulling force exerted on the FA by the stress fibres, and the stiffness of the ECM. FAs are the points of force transmission from the cytoskeleton to the ECM and they change in number, size and composition in response to the level of force experienced [10,11,12,13,14,15,16,17,18].

### 1.2. Focal Adhesions in Health and Disease

Due to their importance in cell adhesion and force transmission, FAs are crucial to most types of cell migration, including in vitro movement on a 2D surface [19]. Migration and adhesion, in turn, are key cellular functions required for many physiological and pathophysiological processes. Of note, they are vital to embryonic development, where FAs coordinate stem cell differentiation in response to ECM stiffness, and they are key to organogenesis and morphogenesis [20,21,22]. The importance of FAs is not limited to foetal physiological processes, as they are also pivotal to the proper functioning of the immune system and to wound healing [23]. Considering their importance in these physiological processes, it is no surprise that FAs also have major roles in many developmental and immunological disorders [19,22,23]. Nevertheless, in pathology, most FA research focusses on their role in cancer, where they are particularly important during metastasis [24,25,26]. Integrins and other FA components are often upregulated in aggressive forms of cancer [27,28,29] and they are crucial to the Epithelial–Mesenchymal Transition (EMT) [30,31,32,33,34].

### 1.3. The Molecular Composition of Focal Adhesions

FAs are flat, elongated structures 1–5 µm long, 300–500 nm wide and, on average, 50 nm thick [35,36,37,38] (Figure 1C). The characteristics of individual FAs, including their molecular composition, varies greatly depending on the different inputs they receive from their local environments, both intracellularly and from the ECM. However, all FAs are bound by actin stress fibres and incorporate integrin transmembrane receptors, heterodimers of α- and β-integrins. Integrins are transmembrane receptors, which are a part of the FA complex and directly bind to the ECM. In mammalian cells, 24 different heterodimers have been reported, all with their own ligand specificity for (sets of) ECM proteins [39].

The intracellular proteins of FA complexes are organised into a layered nanostructure. A seminal study using the single-molecule microscopy technique, three-dimensional interferometric photoactivated localisation microscopy (iPALM, see Section 3.1.3) revealed the presence of three different layers: at the bottom, closest to the adherent membrane (within ~10–20 nm), the integrin signalling layer (ISL), at the top (~50–60 nm from the adherent membrane) the actin-regulatory layer (ARL) and between the force transduction layer (FTL) [37]. The authors demonstrated a consistent stratification of FA proteins in human bone cancer cells (U2OS) and mouse embryonic fibroblasts across FAs of diverse shapes, sizes and maturity levels. This strongly suggested that the layered nature is the result of a cell-type-independent organising principle that persists throughout FA maturation stages. A plethora of subsequent studies employing a range of different techniques, culturing methods and cell types further investigated this layered nanostructure [40,41,42,43,44,45]. The cell-type-independent nature was shown by the diversity of cell types studied, which are from different species (human, mouse and simian) and include both carcinogenic and non-carcinogenic cells. Studied cell types include fibroblasts, mammary epithelial cells, bone cells, endothelial cells and pleuripotent stem cells. The possibility that the layers are artefacts induced by fixation was ruled out by a study that demonstrated FA stratification in living cells [41]. The observed layers are also not an artefact of the relatively complex single molecule sample preparation or image reconstruction processes, since several studies employed variations in the structured illumination microscopy (SIM) technique (see Section 3.1.2) [40,41]. Finally, the layered nature of FAs is supported by studies examining FAs with elegant biochemical approaches, proximity-dependent biotinylation assays [46,47].

The formation of new FA complexes involves several steps (Figure 2). Firstly, integrin transmembrane receptors bind to the extracellular matrix, which induces clustering of the integrins, leading to their activation with resultant conformational changes [12,48,49,50] (Figure 2B). Secondly, more than 30 different intracellular adaptor proteins, including talin and paxillin, are recruited [51,52] (Figure 2C). These proteins, in turn, promote integrin activation, leading to the clustering of more integrins [53]. The adaptor proteins also provide a binding platform for the hundreds of other intracellular FA proteins, which ultimately results in the recruitment of an actin stress fibre (Figure 2D,E). In general, newly forming FAs can be thought of as complexes forming in a layered manner, starting with mainly the bottom layer and a few middle layer proteins, with the actin-interacting proteins added to the top as the complex continues to grow and mature [54]. FAs are typically regarded as highly dynamic complexes. Throughout the cell, FAs are continuously forming, as well as disassembling, with the average FA lifetime being about one hour, depending on several factors, such as migration speed [55,56,57,58].

### 1.4. Focal Adhesions In Vitro and In Vivo

FAs are diverse multi-protein assemblies that vary in their overall structure and protein composition depending on their specific function and environment, with over two hundred different reported proteins [19,59]. These include (trans)membrane receptors, adaptor proteins and many different signalling proteins such as kinases, phosphatases and G-protein regulators. Differences in post-translational modifications by these signalling molecules add significant complexity to FAs [60,61,62]. Moreover, over a thousand proteins are candidates to be added to this list based on mass spectrometry experiments [63,64]. The ECM is also complex; it is a large three-dimensional network, with hundreds of different proteins, proteoglycans and glycosaminoglycans as possible components, that is constantly being reorganised by matrix-degrading enzymes [65]. There is also a direct interplay between the ECM and FAs, i.e., both ECM macromolecular composition as well as its mechanical properties (e.g., stiffness) influence the composition of the FA itself, its size, and its structure [60,66,67].

The combined complexity of FAs, the ECM and their interactions, makes mechanistic studies exceedingly challenging. Consequently, to facilitate mechanistic studies, the complexity of the extracellular environment is often reduced to the coating of a single type of ECM protein on glass. However, concerns have been raised about the relevance of FAs observed on coated coverslips to cell adherence and migration under physiological conditions. The primary concern is the extreme stiffness of glass (over 50 GPa) compared to the stiffness encountered in the human body, which ranges from ~1 kPa for the brain and ~10 kPa for muscle to ~100 kPa for bones. Substrate stiffness has a direct impact on FAs and FA components. For example, if cells are cultured on glass or other stiff substrates, the formed FAs are larger and expression of numerous FA proteins, including talin, paxillin and vinculin, is increased compared to substrates with a stiffness comparable to in vivo conditions [20,66]. However, when cells are plated on gels with a stiffness comparable to in vivo conditions, FAs are still observed. These FAs are composed of the same elements as FAs formed on glass, but they are typically smaller. A second concern is that coated coverslips provide a two-dimensional environment for the cells to bind to, instead of the three-dimensional ECM typically encountered in vivo. Again, this issue has been addressed using gels. Similar to gels with reduced stiffness, FAs are also observed in three-dimensional gels; these are smaller than the FAs formed on glass, but largely composed of the same proteins [60,68]. Therefore, FAs are not only formed when cells are exposed to very stiff, two-dimensional environments. Indeed, FAs have been observed in vivo and recent in vivo studies confirmed their importance for several developmental processes and wound healing [69,70,71,72,73].

## 2. Intracellular Focal Adhesion Proteins

The number of proteins present at FA assemblies is such that it is not feasible to comprehensively review the role of all FA proteins. However, to provide a review that includes as many of the diverse functions served by FA components as possible, we have selected two of the most intensively studied proteins in each of the three layers: bottom layer—paxillin, the crucial adaptor protein essential to FA formation, and Focal Adhesion Kinase (FAK), a principal signalling protein; top layer—zyxin and vasodilator-stimulated phosphoprotein (VASP), which both bind directly to actin; middle layer—talin, one of the very first proteins to bind to clustering integrins of a forming FA complex, and vinculin, which forms an interesting bridge between the three FA layers (Figure 2 and Figure 3).

### 2.1. Bottom Integrin Signalling Layer (ISL) Proteins Paxillin and FAK

In the founding paper by the Waterman–Storer lab, the bottom layer was termed the integrin signalling layer (ISL) [37]. Paxillin is among the most studied of the factors present in this layer. Like several other FA proteins, it binds directly to integrins, specifically interacting with the cytoplasmic tails of several α- and β-integrins [74,75,76,77]. Together with ISL protein kindlin, it is among the first proteins to be recruited to newly forming FAs, where it binds to the clustering integrins [37,51,52] (Figure 2C). The C-terminus of paxillin contains four LIM domains, which are responsible for its FA targeting, particularly the third LIM domain [78] (Figure 3A). Paxillin is one of the proteins with the largest number of potential binding partners within FAs [59]. It has been put forward as the main adaptor protein linking all functional modules of FAs [79], an idea which is supported by the particularly large number of proximal interactions found for paxillin in proximity biotinylation assays [47]. It is a multidomain scaffold protein and one of its main functions within FAs is to recruit and facilitate the efficient interaction of other intracellular FA proteins [80].

Focal Adhesion Kinase (FAK) is a non-receptor tyrosine kinase, which consists of three domains: an N-terminal FERM, which mediates its interaction with membrane proteins, a central kinase domain, and a C-terminal focal adhesion targeting (FAT) domain, which contains binding sites for about 50 FA proteins including paxillin and FTL protein talin [81,82,83] (Figure 3B). FAK is recruited to the forming FA by paxillin, through two binding sites at the N-terminal domain of paxillin. FAK activation requires the autophosphorylation of Tyr397. Next to this, maximal catalytic activity requires the phosphorylation of additional residues by other kinases such as Src [84]. FAK is one the key signalling proteins, through both its own kinase activity and through its scaffolding ability to other signalling proteins. For instance, FAK phosphorylates paxillin at Tyr-31 and Tyr-118 [78,85] (Figure 3A). This phosphorylation creates two Src Homology 2 (SH2)-binding sites, which are the main binding platforms for the other paxillin interactors, among which are many other signalling proteins such as Src [82]. Paxillin phosphorylation by FAK is mechanosensitive, i.e., the increased force experienced by the FA increases phosphorylation levels [18]. The creation of these SH2-binding domains is a key step during FA assembly [86]. It influences FA size, can be used as a measure of integrin signalling and is increased during endothelial–mesenchymal transition (EMT) induced by transforming growth factor β (TGF-β) in various cell lines [33,34,87,88]. Incidentally, top layer protein zyxin is required for the efficient tyrosine phosphorylation of paxillin during TGF-β-induced EMT.

### 2.2. Top Actin Regulatory Layer (ARL) Proteins Zyxin and VASP

Zyxin and vasodilator-stimulated phosphoprotein (VASP) are binding partners that are recruited together to the top layer of forming FA complexes [54,89,90] (Figure 2E). VASP is part of the Ena/VASP protein family, a group of highly related proteins. Note that Ena was named after the Drosophila protein ‘Enabled’. In mammals, the Ena/VASP family consist of three different proteins: VASP, mammalian protein ‘Enabled’ homolog (Mena) and Ena-VASP-like protein (Evl). These proteins are highly related and share the same functional domains. VASP is recruited to FAs in a mechanosensitive manner by zyxin, which is itself targeted to FAs through its C-terminal LIM domains [89,91,92] (Figure 3C).

Actin-binding FA proteins such as zyxin and VASP mediate the regulation of actin by FAs. The zyxin N-terminus contains actin binding sites, but also binding sites for α-actinin, an actin-binding protein that is especially abundant on stress fibres [93,94]. The VASP N-terminus incorporates the WASP homology (WH) 1 domain, also known as the Ena/VASP homology domain 1 (EVH1). This is a protein interactor domain, which binds to a specific proline-rich motif present in zyxin and, in the middle layer, protein vinculin [89,90,95]. These stretches are also binding sites for guanine exchange factors (GEFs) for the small GTPase Rho, through which zyxin, VASP and vinculin, in a co-dependent manner, stimulate actin polymerisation in response to mechanical stimuli [89,91,96,97,98,99,100]. The C-terminus of the Ena/VASP proteins contains the EVH2 domain, which is closely linked to their actin-regulatory functions. The EVH2 domain is composed of three consecutive regions termed blocks A, B and C. The conserved KLKR motif within block A mediates the stimulation of actin polymerisation. Block B contains an F-actin binding site and blocks C terminates in a large alpha-helix. This alpha-helix mediates the tetramerisation of the Ena/VASP proteins, which is required for their efficient functioning [101]. In addition to at FAs, both zyxin and VASP also accumulate at actin–polymerisation complexes, which are periodically distributed along stress fibres [102,103,104]. Zyxin recruits VASP and α-actin to damaged stress fibres to coordinate their repair and maintenance [105].

### 2.3. Middle Force Transduction Layer (FTL) Proteins Talin and Vinculin

Talin is a large adaptor protein that binds directly to β-integrins through its globular head domain, an interaction that stimulates integrin activation [106] (Figure 3D). The talin tail domain binds directly to actin, meaning that talin potentially forms a direct link between the upper and lower FA boundaries [107,108]. Talin binding partners also span the FA layers and include bottom-layer proteins FAK and paxillin, middle-layer protein vinculin and top-layer protein α-actinin. The talin tail domain is rod-shaped and consists of 13 tandem α-helical bundles. It contains nearly a dozen mechanosensitive vinculin binding sites. The binding of vinculin to these sites mediates vinculin activation, also in vivo, and strengthens the link between talin and integrins as well as the link between talin and actin [109,110]. Mammals have two highly homologous talin genes, talin1 and talin2; both isoforms are expressed in most cell types. Inside the cell, talin can adopt multiple conformations, varying between a head-to-tail autoinhibited conformation to an extended mechanically stretched conformation [111]. In the extended conformation, several vinculin biding sites are revealed and vinculin binding to these sites stabilises the extended conformation of talin.

Vinculin is similar to paxillin as it is a large scaffold protein, and it is among the proteins with the most potential interaction partners within FAs [59]. Vinculin is, like paxillin and talin, among the first proteins recruited to newly forming FAs. However, it does not directly bind to the clustering integrins and is consequently recruited slightly later than paxillin or talin [54,59] (Figure 2D). Vinculin, in contrast to paxillin, directly binds actin filaments and is involved in actin regulation at FAs [112]. Vinculin has a head and a tail domain with a flexible linker in between, allowing vinculin to adopt open and closed conformations [113] (Figure 3E). In its closed or inactive form, the head and tail domain interact. When vinculin opens into its active form, several protein-binding sites are revealed. The vinculin head domain shares many important binding partners with the bottom ISL layer protein paxillin, including talin [114]. Together with paxillin and talin, vinculin promotes integrin activation and clustering [112]. Paxillin is required for vinculin’s recruitment to FAs in many, but not all, cell types [18,78,115,116,117]. This process is particularly well-studied in fibroblasts, where the paxillin-mediated recruitment of vinculin is force-dependent and, as such, is influenced by both myosin-II contractility and extracellular matrix stiffness [18]. It requires the tyrosine phosphorylation of paxillin, which is mediated by FAK, another ISL protein, in a mechanosensitive manner. The vinculin tail domain is closely linked to actin regulation. It contains actin binding sites but also binding sites for the ARL-layer proteins α-actinin and the Ena/VASP proteins [118,119,120,121]. Vinculin requires both zyxin and VASP to efficiently stimulate actin polymerisation at FAs [89,96,97,98,99,100].

### 2.4. The Importance of Individual FA Proteins In Vivo

The discussed ISL bottom- and FTL middle-layer proteins are all recruited to FAs at relatively early stages during FA formation, and hence seem to play an important structural role in FAs. Consistent with this idea, knockout mice for any of the highlighted ISL and FTL proteins all suffer lethal deficits early during embryonic development [115,122,123,124]. The examined actin-regulatory-layer proteins are recruited at much later stages, which would suggest that the structural importance of these FA proteins is limited. This is supported by the observations that zyxin and VASP knockout mice are viable [125,126].

To gain further knowledge of the role of the ISL bottom- and FTL middle-layer proteins, despite the embryonic lethality of the full knockout mice, cells were cultured from embryonic knockout mice or conditional ‘knockout’ mice were used. Fibroblasts cultured from paxillin (bottom layer) knockout mice display aberrant FAs, decreased migration and problems with cell spreading. Conditional ‘knockout’ mice revealed that FAK is particularly important to angiogenesis and vascular development, and endothelial cells of these mice also display decreased migration [127]. Induced vinculin ‘knockout’ embryonic fibroblasts display problems with cell spreading, with adhesion to a variety of different substrates, three-dimensional invasion, FA maturation and the formation of strong traction forces at FAs, although their random migration velocity is increased [128,129,130,131].

Although viable, the study of VASP-depleted or VASP knockout cells or knockout mice is complicated by the presence of the other Ena/VASP family members and the resulting potential redundancy of (a part of) VASPs functions. Nevertheless, a recent study using somatic gene disruptions of all three family members showed that the Ena/VASP proteins positively contribute to cell adhesion and cell migration over two-dimensional stiff surfaces [132]. Interestingly, zyxin knockout mice display no lethal embryological developmental problems or obvious histological abnormalities [125,133]. However, loss of zyxin, VASP or their interaction, results in cells becoming unable to remodel their cytoskeleton in response to internal or external cues. Such cells are no longer able to thicken their stress fibres in response to mechanical stress or the actin stabilizer jasplakinolide [96,97,134]. This translates into inflexible cellular behaviour. Fibroblasts cultured from zyxin knockout mice are unable to adjust their migratory speed or adhesiveness in response to cues from the ECM, although, overall, both migration and adhesiveness are enhanced in these cells compared to wild-type, suggesting a regulatory, moderating function of zyxin in these processes. The TGF-β-induced EMT response involves the upregulation of zyxin, where zyxin coordinates the remodelling of the actin cytoskeleton during EMT [33]. In line with this important role of zyxin in the EMT process, zyxin has been strongly linked to several types of cancer (progression), including bladder and breast cancer and Ewing’s sarcoma, where it acts as a tumour-suppressor [135,136,137].

### 2.5. FA Proteins and Regulation of Gene Expression

In response to mechanical stimuli, FAs influence processes in the cell nucleus and several of the highlighted FA proteins shuttle between FAs and the nucleus. The C-termini of both paxillin and zyxin contain several double zinc-finger LIM domains, reminiscent of zinc fingers of transcription factors [78,138]. Indeed, both proteins are transcription factors, have nuclear export signals and shuttle between FAs and the nucleus, where they promote gene expression [133,139,140,141]. Apart from in the nucleus, both zyxin and paxillin also influence gene expression at the translational level in the cytoplasm. Zyxin is a direct polyadenylate-binding protein, while paxillin interacts with polyadenylate-binding protein 1 (PABP1), both in the endoplasmic reticulum and at FAs [142,143,144]. It is currently unclear how exactly these proteins translocate to the nucleus, as they do not harbour a canonical nuclear localisation signal (NLS) peptide, but it has been suggested that they are transported to the nucleus by other proteins that do have an NLS [133]. The translocation of paxillin to the nucleus is inhibited by its interaction with vinculin at FAs [145]. Nuclear zyxin is also involved in the control of mitosis progression [104]. FAK is another FA protein, which shuttles between FAs and the nucleus [146]. The N-terminal FERM domain of FAK incorporates a classic nuclear localisation sequence, and both its FERM and kinase domains contain a nuclear export signal, the second of which seems to be the most important [147,148]. Nuclear FAK stimulates cell proliferation and survival by promoting the degradation of p53 through enhancement of its ubiquitination [146]. FAK also influences gene expression by interacting with various transcription factors [81].

## 3. Advanced Light Microscopy Techniques

To gain further knowledge of the interaction of individual FA proteins and focal adhesions and the internal organisation of the proteins within the FA complex, advanced light microscopy techniques have proved to be especially useful, although the resolution of light microscopy is limited. Modern advanced microscopy methods are focused on ways to break, or, more accurately, circumvent, the resolution limit, including total internal reflection (TIRF), structured illumination (SIM) and single-molecule imaging techniques such as photoactivated localization microscopy (PALM) or stochastic optical reconstruction microscopy (STORM). Other advanced imaging techniques focus on using microscopy to obtain dynamic information about the behaviour of individual proteins from the images, information beyond the location of proteins. Examples of such techniques include fluorescence recovery after photobleaching (FRAP), fluorescence correlation spectroscopy (FCS) and spatio–temporal image correlation spectroscopy (STICS), which focus on obtaining quantitative measurements related to the dynamic behaviour of proteins.

### 3.1. Advanced Light Microscopy Techniques Circumventing the Resolution Limit

#### 3.1.1. Total Internal Reflection Microscopy

TIRF microscopy breaks the resolution limit in one direction by changing the angle of the excitation laser. It was first developed in the early 1980s [149], then adapted for easy use in cell biology in the late 1980s [150]. In TIRF the excitation laser illuminates the sample at an angle. Consequently, the laser is reflected at the glass–fluid interphase (Figure 4A). The reflecting laser light creates an electromagnetic field, the evanescent wave. As the wave exponentially declines, it is only strong enough to excite fluorophores up to a very small distance from the glass–fluid interphase, about 100–200 nm. Therefore, FAs are especially suited for TIRF imaging, as most FA components, apart from their enrichment at FAs, are highly expressed in the cytoplasm (Figure 4B).

#### 3.1.2. Structured Illumination Microscopy

Structured Illumination Microscopy (SIM) also uses a specialised form of illumination to improve resolution. While in TIRF only the z-resolution is improved, SIM circumvents the resolution limit in all three directions [151]. In each direction, the resolution is improved approximately two-fold compared to a confocal microscope, in essence enabling the visualisation of three-dimensional objects eight (2^3^) times as small as conventional light microscopy [152].

SIM makes use of a grated light pattern, which produces the Moiré effect (Figure 5A). Images are reconstructed through Fourier transformations, on the basis of the illumination pattern and the Moiré pattern in the raw data [153] (Figure 5A,B). The reconstruction of SIM super-resolution images requires several raw images. Three to five rotations of the illumination pattern are required, and the illumination pattern needs to be shifted to cover the whole image. However, because SIM is a widefield microscopy technique, it is still fast enough for live-cell imaging.

#### 3.1.3. Single-Molecule Microscopy

Single-molecule microscopy makes use of fluorescent markers that switch between two states in a light-controlled manner. Essential to single molecule microscopy is the ability to limit these transitions to only a few fluorescent emitters at a time, allowing for the collection of signals from single emitters, the localisation of which can then be determined with a high accuracy of from ~5 to 30 nm [154,155] (Figure 6). Two main types of single-molecule localisation microscopy are distinguished based on the type of fluorescent markers used: in stochastic optical reconstruction microscopy (STORM) fluorescent organic dyes switch between light and dark states, while in PALM, fluorescent proteins switch between light and dark states (photoactivatable proteins) or between different colours (photoswitchable or photoconvertible proteins) (see also Section 3.2.2).

### 3.2. Advanced Imaging Assays to Study Protein Dynamics

#### 3.2.1. Fluorescence Recovery after Photobleaching

Fluorescence Recovery After Photobleaching (FRAP) is used to obtain quantitative parameters of the mobility and dynamics of proteins. In a typical FRAP experiment, a small region of the cell is briefly illuminated with a powerful laser, bleaching the fluorescent tags of the fusion proteins. After bleaching, fluorescence levels are recorded and plotted against time in FRAP-curves.

A lot of information can be extracted from FRAP-curves about the mobility and binding dynamics of the studied proteins, especially when the experimental FRAP-curves are further analysed by fitting them to curves generated by Monte-Carlo-based simulations [157]. The obtained quantitative parameters may include the number of fractions with distinct dynamic parameters that are present within the protein pool, the sizes of these fractions and the associated on-rate (k_on_)—and off-rate (k_off_) (the rates at which the proteins bind to and dissociate from the focal adhesion complex), while avoiding most of the simplification steps needed for FRAP-curve analysis using mathematical equations [158,159].

#### 3.2.2. Photoactivation and Photoswitching

The optimisation of fluorescent proteins also led to the development of proteins with altered emission characteristics, due to photochemical reactions or through conversions between chromophore stereoisomers [160], in response to the light of certain wavelengths, and the photoactivatable and photoswitchable or photoconvertible fluorescent proteins [161]. Photoactivatable proteins require activation by illumination at a wavelength shorter than their excitation wavelength to become efficiently fluorescent [162]. Continuing development has led to the engineering of a large number of photoactivatable fluorescent proteins with differing characteristics, for example, in the emission spectrum, in the brightness and in the reversibility of the activation [163].

For the photoswitchable or photoconvertible fluorescent proteins, irradiation with laser light of roughly four hundred nanometres results in a shift in their excitation and emission spectra. Continuing research has led to the development of a large number of photoconvertible fluorescent proteins, which together span a large emission spectrum with varying levels of brightness and reversibility [163].

A common application of photoswitchable and photoactivatable proteins is in the super-resolution techniques such as Photo Activated Location Microscopy (PALM) and Stochastic Optical Reconstruction Microscopy (STORM) [164,165,166] (see also Section 3.1.3). Another, much simpler application of photoconvertible proteins is an assay that specifically visualises the stably bound fraction of a protein (Figure 7) [58]. While traditional FRAP provides accurate estimates of the size of the stably bound fraction, the photo conversion assay reveals the spatial distribution of the proteins of this fraction within a macromolecular complex, distinguishing it from its dynamically exchanging counterpart. A particular advantage of this new technique is that the stably associated proteins are directly visualised and the complex remains visible throughout the experiment, which is especially useful when investigating the association of proteins with dynamic macromolecular complexes such as FAs.

### 3.3. Advances in Data Analysis

Large improvements in the employed data analysis methods also contributed to the increased importance of light microscopy to modern biology, as well as all the technological advances involving light microscopes and fluorophores. Most notably, the single-molecule techniques of PALM and STORM depend entirely on post-acquisition data analysis techniques to create super-resolution images. Similarly for SIM, elaborate data analysis, involving Fourier transformations, is required to reconstruct high-resolution SIM images from the raw data. Due to this heavy reliance on data analysis techniques, the widespread availability of advanced data analysis techniques also became necessary and the interest in advanced data analysis has increased. With increased availability and interest, it is only natural that the image analysis of data acquired through other imaging techniques also improves, allowing for more data to be extracted from these images. A good example of this is the increased use of deconvolution software [167]. The availability of free open-source software such as ImageJ [168] and its framework Fiji [169] has allowed researchers to develop their own image-processing or visualisation techniques or to adapt and build upon the image analysis methods developed by others. The improvements in quantification, which increased the potential for reliable comparison between different conditions, has contributed strongly to many important scientific advances in the field of biology [170].

## 4. Focal Adhesion Protein Dynamics

As discussed above, the advances in light microscopy, particularly super-resolution, revealed the layered structure of FAs. Subsequently, the possibility of assessing the dynamic properties of individual FA proteins and their interplay with the layered FA complex in living cells by functional imaging methods such as FRAP has contributed largely to the current knowledge of FA function. The application of a variety of FRAP assays revealed that, similar to many other cellular structures, the FA components show very dynamic interactions with the FA complex [18,58,62,68,141,145,171,172,173,174,175,176,177,178,179,180,181]. These studies provide useful insights into the factors that influence the binding dynamics of proteins at FAs (Figure 8).

### 4.1. FAK Activity and the Dynamics of Focal Adhesion Protein

The activity levels of signalling proteins can influence FA protein dynamics (Figure 8, item 1). The activity of the FA key signalling factor FAK has been demonstrated to influence paxillin and vinculin dynamics, although there is a discrepancy between studies regarding the nature of its effects. One study examining tyrosine phosphatase-2 (Shp2) knockout cells, cells in which FAK is hyperactivated because Shp2 inhibits FAK activity by dephosphorylation of its activating pTyr397, demonstrated a faster recovery of paxillin and vinculin [171]. The authors also showed the rescue of the dynamics by FAK inhibitors. Conversely, another study using a FAK inhibitor found no significant effect on vinculin dynamics, while paxillin recovery was again faster [177]. Additionally, the authors observed a faster recovery of an artificial construct containing only the SH2 domain, which, after phosphorylation by tyrosine kinases such as FAK, acquires protein-binding properties. This suggested that the faster recovery of paxillin upon FAK inhibition was mediated by its SH2 domain. The inconsistencies between the studies, with the overactivation as well as inhibition of FAK activity resulting in faster paxillin dynamics, might be due to the different culturing conditions between studies. However, it may be that paxillin dynamics require carefully regulated FAK activity levels, and disruption of this regulation through either inhibition or hyperactivation both result in increased paxillin dynamics.

### 4.2. The Effects of Force

The level of force experienced by FAs, determined by the balance between the force transmitted through the attached actin fibre as a result of myosin-II contractility and ECM compliancy (stiffness), is another factor shown to influence FA protein dynamics (Figure 8, item 5).The level of force transferred through actin to FAs has also been demonstrated to influence paxillin, zyxin and vinculin dynamics; however, again, the nature of this effect was different between studies. Reducing the force by pharmacological inhibition (blebbistatin or Rho-inhibitor Y-27632) of the contraction of myosin-II increased the stably bound fractions of paxillin and zyxin (bottom and top layer, respectively), but had the opposite effect on vinculin (middle layer) dynamics in different studies [62,178]. For vinculin, this was replicated in a study which used substrate compliancy (stiffness) to influence the force experienced by FAs instead of altering myosin-II contractility [175]. An increased force on FAs correlated with an increase in the stably bound fraction of vinculin. A fourth study [68], however, found no effect of force on vinculin dynamics. Furthermore, in contrast to the first two studies, the authors found that decreasing the force through decreased substrate stiffness correlated with faster zyxin recovery and a decrease in its stably bound fraction. One noteworthy difference in this study, potentially contributing to its opposite findings, is that cells were studied in a 3D environment. In a 3D environment, where cells are surrounded by ECM and can form FAs throughout their plasma membrane, the forces transmitted to individual FAs might be different than in a 2D environment, where FAs are only formed at the ventral plasma membrane.

### 4.3. The Effect of Protein Conformation and FA Size and Maturity

Several other factors have been shown to influence vinculin-binding dynamics [173,175]. For example, vinculin mutations that inhibit the interaction between its head and tail domains, forcing it into its open conformation, were shown to significantly slow its speed of recovery and increase its stably bound fraction [173]. This was partly mediated by vinculin–talin interactions, since an additional mutation of the talin-binding site (which is only exposed in the open form) in the vinculin head domain partially rescued wild-type vinculin dynamics. The presence of open vinculin mutants also slowed down talin dynamics (measured through recovery half-times), but had no effect on the dynamics of paxillin and α-actinin, which also interact with vinculin. On a cellular scale, FA maturation and cell movement have also been demonstrated to correlate with vinculin dynamics [175]. Its recovery was much slower, and its stably bound fractions were much higher in the mature, and therefore larger, FAs found in stationary cells than in small FAs in actively migrating cells. Similarly, slower paxillin and FAK dynamics at larger FAs have also been reported [181]. Taken together, it seems that small FAs in fast-migrating cells are more dynamic than the larger FAs typical of stationary cells, which is consistent with the need for transient cellular adhesion in migrating cells versus a more stable connection with the ECM in stationary cells. Furthermore, for vinculin, its slower dynamics at larger FAs would suggest that, at these FAs, a higher fraction of vinculin molecules is found in the open, active, conformation, promoting interaction with talin. Together, these studies indicate that FA protein-binding dynamics correlate with the size and maturity of the FA (Figure 8, item 2), as well as with the conformation of its proteins (Figure 8, item 3).

### 4.4. Endosomes and Paxillin Dynamics

Finally, endosomes, which are targeted to FAs along microtubules, were shown to influence paxillin dynamics. Preventing the formation of the large scaffold complex characteristic of late endosomes strongly increased the stably bound fraction of paxillin [179]. Likewise, the absence of stonin1, a clathrin-adaptor specific to the endocytosis of the integrin co-receptor NG2, has been demonstrated to strongly increase the stably bound fraction of paxillin, an effect which was rescued by stonin1 expression [180]. This makes endosomes another factor demonstrated to influence FA protein-binding dynamics (Figure 8, item 4).

### 4.5. Relative Dynamics of FA Proteins

Fruitful direct comparisons between the different FRAP studies on FA protein dynamics [18,58,62,68,141,145,171,172,173,174,175,176,177,178,179,180,181] are complicated by the large variation in reported quantitative parameters between studies. For instance, for paxillin in fibroblasts, by far the most examined combination of protein and cell type, the half times to full recovery varied between 1.5 and 41 s, the times until final recovery between 30 and 200 s, and mobile fractions between 60% and nearly 100%. The period fibroblasts were cultured on fibronectin prior to imaging and ranged from 15 min to 48 h, which influences the level of spreading, a factor known to affect FA maturation that has been shown to inhibit vinculin dynamics [175]. Moreover, different culturing conditions may lead to differences in FA composition and the phosphorylation and or conformational states of various FA proteins, all of which can influence protein dynamics [18,62,141,145,171,174,175,177].

Despite the large differences between studies, by comparing the dynamics of different proteins examined within the same study, keeping culturing and analysis methods consistent, it is possible to examine their relative dynamics. One study examined all proteins highlighted in this review: this study found slow dynamics of the integrin signalling (bottom)-layer proteins (paxillin, FAK) and the force transduction (middle)-layer proteins (vinculin, talin), but much faster dynamics of the actin regulatory (top)-layer proteins (zyxin, VASP) [62]. Conversely, zyxin and VASP had very small stably bound fractions, talin and FAK had intermediate stably bound fractions and vinculin and paxillin had much larger stably bound fractions. These patterns of relative recovery speeds and stably bound fraction sizes have largely been replicated in other studies [18,58,177,178,181], examining different cell types from different species. One study showed slower recoveries and larger stably bound fractions for paxillin and vinculin compared to zyxin [178]. This was replicated in another study, which additionally described the dynamics of VASP [58]. Here, VASP showed the fastest recovery, closely followed by zyxin, while paxillin and vinculin displayed much slower and remarkably similar recoveries. The stably bound fraction was very small for VASP, intermediate for zyxin and very large and remarkably similar for paxillin and vinculin. A third study also found such slower dynamics for vinculin, paxillin and talin, but both zyxin and FAK were found to recover much faster [18]. This faster recovery of FAK compared to vinculin and or paxillin was replicated in two other studies [177,181].

FAs are typically regarded as highly dynamic complexes, as they show a sliding type of movement [182,183] in both migrating and stationary cells. However, a study using Monte-Carlo-based simulations (see Section 3.2.1) to quantify the binding dynamics of paxillin, vinculin, zyxin and VASP [58], revealed that, for all four proteins, next to dynamically exchanging pools (interactions ~1 min or less), a pool with stable associations (>30 min) was present at FAs. The average residence time of these stably associated fractions was relatively long compared to FA lifetimes [55,56,57], indicating that the proteins in these fractions (which for paxillin and vinculin is almost 50%) remain associated for a large part of the lifetime of a FA and revealing some properties of FAs are less dynamic than previously thought.

Overall, the relative dynamics between the different proteins (slower recovery and larger stably bound fractions for the bottom and intermediate layer proteins, faster recovery and smaller stably bound fractions for the top layer proteins) were largely consistent between studies, despite the differences in the precise reported dynamic parameters or the different cell types examined, suggesting these patterns are a general property of FAs. This might be a result of the different functions served by the proteins present in the different layers. Many of the bottom and intermediate layer proteins are recruited early during the formation of novel FA complexes and have many different potential interaction partners, suggesting that they have a more structural role within the FA complex. More stable interactions of such structural proteins might be required for the overall stability of the FA complex. The top layer proteins, however, are recruited much later and seem to have much less of a structural role within the FA complex; instead, they are primarily important for the regulation of attached actin stress fibres. For proteins responsible for the regulation of such a fast-changing network, more dynamic interactions might be advantageous to allow fast changes in regulation in response to changes to the network.

### 4.6. FA Orientation and Location and Its Protein Dynamics

The location and orientation of FAs within the cytoplasm also has been shown to be correlated with different dynamic properties. To investigate this in cells, FAs can be classified based on their angle (Figure 8, item 6) and distance from (Figure 8, item 7) the closest edge of the ventral part of the plasma membrane, i.e., the part adhered to the glass cover slip [58]. The correlation between FA location and orientation and its dynamics was mostly observed for zyxin and VASP, and most especially for FAs located close to the nearest ventral membrane edge and orientated with their long axis perpendicular to it. At these FAs, zyxin and VASP (which bind actin) were observed to interact with FAs in a significantly more dynamic manner, potentially to facilitate the coupling of actin to these FAs. Strong links between actin and FAs that lie close and perpendicular to the edge of the ventral membrane may be needed to generate the force required to protrude or retract the ventral membrane. The reported effects of FA location and orientation on paxillin and vinculin dynamics are more subtle, but it is noteworthy that, at FAs close to the membrane edge, paxillin-binding was significantly more dynamic, presumably stimulating the dynamics of these FAs since paxillin is a key structural component. This is supported by the finding of increased on-rate constants of the dynamically bound VASP/zyxin fractions at FAs close to the membrane edge. Additionally, another study found that mutations of Rab40b, which increased FA stability, correlated with an increase in the fraction of FAs further from the membrane edge compared to FAs close to this edge [184]. This could be advantageous for FAs at the edge of the ventral membrane to be more dynamic when a cell is exploring its immediate environment by protruding its membrane at different areas but is not yet committed to moving in a particular direction. Interestingly, another study points towards the biological consequences of disruption of the ratio of FAs that are far from, compared to close to, the adherent membrane edge, as this ratio was increased in fibroblasts from individuals with bi-allelic pathogenetic variants in an enzyme that is important to the synthesis of heparan sulfate proteoglycans, important signalling molecules associated with the plasma membrane or found in the ECM [185]. The observed location- and orientation-dependent dynamics may be regulated by several factors that form gradients based on their distance from the ventral membrane edge, such as actin fibre thickness and connectivity, and the concentration of (signalling) molecules and enzyme activity [186,187,188,189,190,191,192,193], effectively creating different local environments for FAs that vary with their distance from the edge of the ventral membrane.

### 4.7. Paxillin and Vinculin versus Zyxin and VASP—A Recurring Theme

Overall, when comparing the binding dynamics of the different FA proteins, there seems to be a clear split between zyxin/VASP (top layer) versus paxillin/vinculin (bottom- and middle layers, respectively) [18,58,62,178]. Zyxin and VASP show similar fast dynamics with small stably bound fractions that strongly correlate with FA location and orientation (see Section 4.6). Conversely, paxillin and vinculin show much slower dynamics with a large stably bound fraction, on which the impact of FA location or orientation is smaller. Interestingly, in studies investigating different aspects of protein behaviour at FAs, a split between paxillin/vinculin versus zyxin/VASP was also observed. For instance, top-layer zyxin/VASP dissociate from disassembling FAs earlier than paxillin/vinculin, including in response to actomyosin-II inhibition [62,102]. When stress fibres thicken, in response to mechanical stress or to the actin stabilizer jasplakinolide, zyxin/VASP rapidly translocate from FAs to the thickening stress fibres, while vinculin/paxillin remain associated [96,134]. Zyxin is completely lost from FAs in response to actin polymerisation inhibition, while vinculin levels remain unchanged [141]. Overexpression of the zyxin LIM-domain causes the loss of endogenous zyxin and VASP from FAs, while vinculin levels remain unchanged [91,99]. Since paxillin and vinculin are both large structural adaptor proteins with many binding partners, many of which are shared, including each other, slower and similar dynamics might arguably be expected for these proteins. On the other hand, vinculin also shares many binding partners with zyxin and VASP, such as actin and zyxin/VASP themselves [89,96,97,98,99,100,112,118,119,120,121]. Therefore, the split cannot be explained by an abundance of shared binding partners alone. Instead, a strong possible explanation is found in the layered architecture of the FA complex: zyxin and VASP are both found in the top ARL layer while vinculin and paxillin are observed in the middle and bottom layer, respectively. An overall distinction between paxillin/vinculin and zyxin/VASP behaviour is a recurring theme, indicating that the layered architecture of FAs is mechanistically relevant.

## 5. Protein Organisation within FAs

### 5.1. The Distribution of Focal Adhesion Proteins at the Nanoscale

The application of super-resolution advanced microscopy techniques in FA research revealed their internal structural protein organisation at the nanoscale, which includes the previously discussed layered structure. However, the internal structure of FAs is not as well-defined as, for example, that of podosomes [194,195], protein complexes analogous to FAs that link the actin cytoskeleton to the ECM in certain blood and cancer cells and specialize in the degradation of the ECM, perhaps due to the higher number of different proteins localised to FAs. Nevertheless, several studies examining FAs at the nanoscale demonstrated other non-homogenous protein distributions at the nanoscale (Figure 9A). Several studies using single-molecule (see Section 3.1.3 and Figure 6) or electron microscopy techniques demonstrated proteins aggregating into clusters of ~25–100 nm within FAs in both fixed and live cells [196,197,198,199,200,201]. Studies using SIM (see Section 3.1.2 and Figure 5)-based techniques additionally found subdomains along the entire length of the longitudinal FA-axis that are approximately 300 nm wide [202,203]. However, in these studies, cells were imaged only 3 or 4 h post-seeding, suggesting that this might be a feature specific to less mature FAs in spreading cells, as such a distribution was not seen in a similar study examining fully spread cells 48 h post-seeding [199]. Indeed, another study examining FAs during the process of cell spread observed that FAs are initially composed of 300 nm wide subunits, which subsequently split during FA maturation as spreading proceeded [204]. Additionally, studies using SIM-based techniques demonstrated the accumulation of two talin-binding adaptor proteins, kank1 and kank2, specifically around the FA edges [205,206]. Finally, in especially large and stable FAs unique to pluripotent stem cells at colony edges, termed cornerstone FAs, β5-integrins and talin were enriched around the FA edges and kank2 accumulated specifically around the distal FA end [42].

Additionally, results from a recent study suggests the FA layers are shifted relative to each other along the FA in a hepatocyte growth factor (HGF)-dependent manner [207] (Figure 9B). Paxillin (ISL), vinculin (FTL) and zyxin and VASP (ARL) were used in various combinations of two as markers for the layers and studied by SIM. In the most frequently observed configuration, the ISL (bottom) layer protruded at the FA head (the FA end closest to the edge of the adherent membrane). At the FA tail (the FA end where the stress fibre enters), zyxin and VASP from the ARL (top) and vinculin from the FTL (middle) extended further, with vinculin from the FTL forming the tail tip (Figure 9B). Of note, talin was shown to be significantly more stretched and extended at FA tails compared to elsewhere in the FA [37,43], and talin-stretching unmasks cryptic vinculin binding sites on talin [110]. Interestingly, HGF-induced scattering, which reduces the contact between cells and increases undirected migration [208,209,210,211], altered the observed layer shifts [207], where paxillin heads were shorter and zyxin tails longer than in unstimulated cells. Moreover, FAs at protruding or retracting membrane edges had longer paxillin heads than FAs at static edges. Taken together, this suggests that the repositioning of layers along FAs plays an important role in both overall cell movement and in the movement of the cell membrane.

### 5.2. Heterogeneous Distribution of Protein Activity or Binding Dynamics within FAs

Apart from their heterogeneous internal distributions, the binding dynamics or activity levels of FA proteins were also shown to correlate with their localisation within FAs (Figure 9A). For example, active and inactive integrins were shown to form discrete nanoclusters within FAs [203]. The binding dynamics of paxillin were demonstrated to vary along large FAs at the trailing end of actively migrating cells [176,212,213]. Furthermore, long-term immobile or stably bound paxillin and vinculin were shown to be specifically concentrated into discrete areas within FAs [58]. These concentrated areas of stably bound proteins were most often located close to the FA tails, which is in line with studies that showed that the inactive form of vinculin is enriched at FA heads and the active form at FA tails [43,214], since active vinculin is known to be more stably associated [18,173]. Finally, the stably bound paxillin was concentrated in small clusters and vinculin is more dispersed [58]. This difference in concentration might be because paxillin directly binds to integrins, which, during the early phases of FA complex formation, strongly cluster together, with clustering growing even more pronounced for activated integrins [203]. Conversely, vinculin directly binds to the force-bearing F-actin, potentially pulling the vinculin clusters further apart.

### 5.3. FA Proteins Are Not Strictly Separated into Layers

Despite the numerous different ways in which the layered structure of FAs is demonstrated, it is also important to realise that these layers may partly overlap and specific proteins may be enriched in a specific layer rather than strictly present in only one layer. For example, paxillin, as a direct integrin-binding protein, is considered an ISL bottom-layer protein. In all studies examining paxillin height within FAs, its average z-positions fall within the bottom layer and the majority of paxillin associated with FAs is located in this layer [37,41,42,43,44]. However, about half of the paxillin proteins were observed within the ISL, with the remaining half spread over the two higher FA layers [43]. Similarly, when examining the height of actin itself, an actin-regulatory layer protein, only about 75% of the actin at FAs was observed at the height of this top layer and the remaining 25% was found at lower positions within the FA complex. Therefore, while the layered nature of FAs is seen very consistently, there is a degree of overlap between the layers, and the definition of their boundaries is partially arbitrary and, therefore, may not be accurate for all FAs.

Furthermore, for some proteins, there is a level of fluidity to their height within the FA complex, and sometimes even to the FA-layer to which they localise. For example, although its average z-position continued to fall within the bottom ISL layer, in motile cells, paxillin was observed at significantly lower positions in maturing FAs at retracting regions than at stable FAs in non-retracting regions, or than in small newly formed FAs at protruding regions [41].

Proteins of the middle force transduction layer seem to have a unique position within FAs. Different studies found the FTL-layer protein talin to be preferentially arranged in a tail-above-head (downward) orientation in mature FAs, with its N-terminus overlapping with proteins of the bottom ISL layer and its C-terminus overlapping with proteins of the top ARL layer [37,40,41,42,43]. The observed contact angle between the talin molecules and the integrins in the plasma membrane was ~15°, a parameter which was consistent between cell types, the majority of the talin proteins at FAs were also consistently extended, with limited unfolding of some domains [40,42]. The talin molecules diagonally spanning the FA layers seem to serve a critical role in determining the position of the FA layers [40]. Without talin, no FAs are formed, but FAs with a stratified nanostructure were induced by the expression of progressively shorter talin analogues that maintain the N-terminal FERM domain responsible for integrin interaction [106] and the C-terminal TATCH domain responsible for actin binding [40,106,107]. The z-position of both the actin stress fibres and top ARL protein VASP, as well as of intermediate-layer protein vinculin, were all positively correlated with the length of the expressed talin analogues. This suggests that the physical length of the talin proteins determines the vertical position of the different FA layers.

The position of vinculin, another FTL middle-layer protein, has also been demonstrated to be tightly regulated within the layered FA nanostructure. The largest fraction of vinculin proteins at FAs is generally observed in the middle layer, but various factors have been demonstrated to influence this [37,43]. At mature FAs, the majority of vinculin proteins were shown to be in the active open conformation [214]; like talin proteins, these open vinculin molecules were shown to adopt a downward, tail-above-head, position [43,215]. Activated talin molecules were primarily observed in the middle FTL layer, but a significant fraction was also found in the top ARL layer, with little activated vinculin observed in the bottom ISL layer [43]. Inactive vinculin molecules, adopting a closed conformation with the vinculin head and tail domains interacting, were primarily seen in the bottom ISL layer, with a significant fraction in the middle FTL layer, but little inactive vinculin was found in the top ARL layer. Vinculin talin interaction was demonstrated to be required for the activation of vinculin at FAs, increasing the average height of activated vinculin at FAs, and shifting a proportion of active vinculin molecules from the intermediate FTL to the top ARL layer [43]. Furthermore, the lowering of the average position of inactive vinculin at FAs, shifting a larger proportion of these proteins to the bottom ISL layer, was demonstrated to require interaction with paxillin phosphorylated at tyrosines 31 and 118 by FAK.

Two very recent studies point towards the biological importance of preserving the layered nanostructure of FAs [44,45]. The first study revealed that the chlamydial protein Translocated actin-recruiting Phosphoprotein (TarP) is targeted to FAs by vinculin, where it increased FA stability and inhibited cell motility [44]. This has important clinical consequences, because when the attachment of cells to the ECM is enhanced,, this inhibits extrusion, a mucosal epithelial defense mechanism against infection in which infected cells are released from the epithelium to prevent the infection from spreading. The authors were also able to demonstrate that TarP displaces FAK and paxillin from their normal positions within the bottom ISL layer to the top ARL layer. Similarly, the second study showed that the expression of a kindlin mutant, which is shifted from the bottom ISL layer into higher FA layers, also has severe biological consequences, with strongly attenuated cell-spreading and reduced FA formation [39]. Kindlin is another important integrin-activator, such as talin, and kindlin and talin often work together in a synergistic manner for the efficient activation of integrins [216,217,218]. Restoring the ISL-targeting of the kindlin mutant by adding a non-specific lipid anchor to target it towards the plasma membrane, rescued both the cell spreading and the FA formation deficits. While, in both studies, the alteration in the layered nanostructure cannot be directly linked to the observed attenuation of FA formation and cell spreading, it is certainly tempting to speculate.

## 6. Conclusions

It has become increasingly clear that focal adhesions are internally organised into different smaller and larger subdomains with different compositions, with regard to both their protein components, as well as the activity or binding properties of these components. An important aspect of this organisation is the layered nanoarchitecture of FAs, a general feature of FAs that is conserved across cell types and species. Furthermore, the adoption of a layered view appears to aid in the understanding of various aspects of FA protein behaviour, as we have highlighted in this review. A good example of this is the relative binding dynamics of FA proteins, another characteristic that remains consistent across cell types and species. That is, proteins of the bottom two layers (ISL and FTL) are considerably less dynamic than proteins of the top layer (ARL). However, biological systems rarely conform to overly simplistic models, and FA stratification is no exception. FAs are complex and diverse cellular structures that respond to a wide range of stimuli, leading to considerable heterogeneity between individual FAs, as they respond to diverse stimuli, including the precise nature of their layered architecture. Proteins are enriched in specific layers rather than uniquely localised to a single layer; proteins can shift between layers in response to their activity levels and at least two proteins of the force transduction layer, vinculin and talin, are orientated almost perpendicular to the layers spanning the entire layered architecture of FAs. Taking all this into consideration, to understand the internal structure of FAs, it is probably most helpful to visualise a structure composed of three gradients located on top of each other rather than three discrete layers. Despite these considerations, the strong conservation of the layered nature of FAs across cell types and species suggests that it is more than an interesting structural feature of FAs and has regulatory relevance.

## Figures and Tables

**Figure 1 biology-10-01189-f001:**
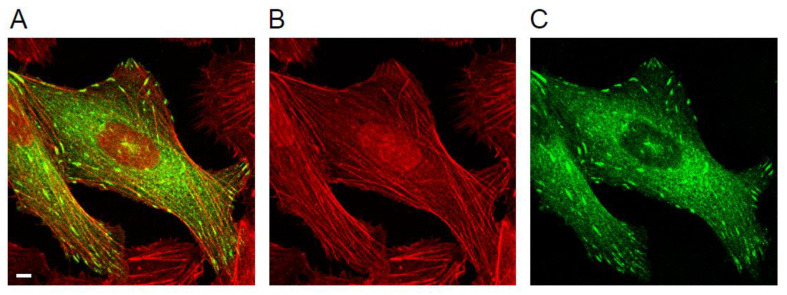
Focal adhesions (FAs) and stress fibres. (**A**) Overlay of the maximum projection of the confocal image of a U2OS cell stably expressing the intracellular FA protein paxillin-GFP (green) and stained with phalloidin-CF405 (pseudocolour red), a toxin that specifically binds to F-actin such as the stress fibres connecting to FAs. Note that the other FA proteins discussed here show identical distributions at confocal resolution. Scale bar: 5 µm (**B**) phalloidin channel of the data shown in (**A**). (**C**) GFP-channel of the data shown in (**A**).

**Figure 2 biology-10-01189-f002:**
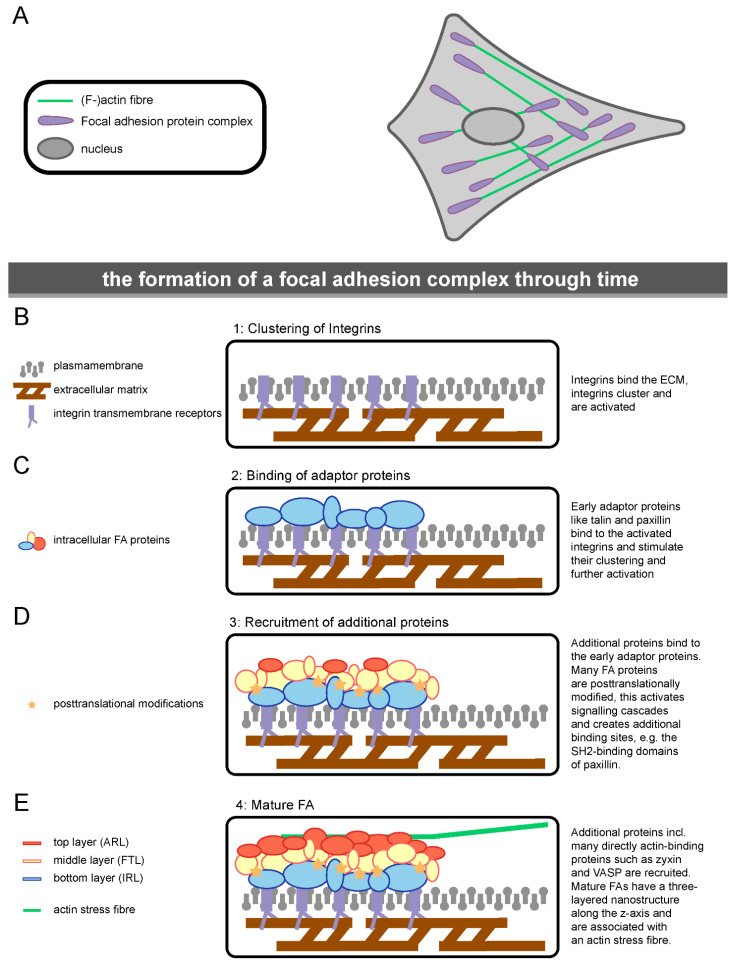
The formation of a focal adhesion complex through time. (**A**) Schematic representation of FAs in a cell (not to scale). (**B**–**E**) Simplified overview of the main steps involved in the formation of a focal adhesion complex. Steps shown in chronological order, from the initial clustering of integrin receptors (**A**) to the eventual formation of a mature three-layered focal adhesion with associated F-actin stress fibre (**D**).

**Figure 3 biology-10-01189-f003:**
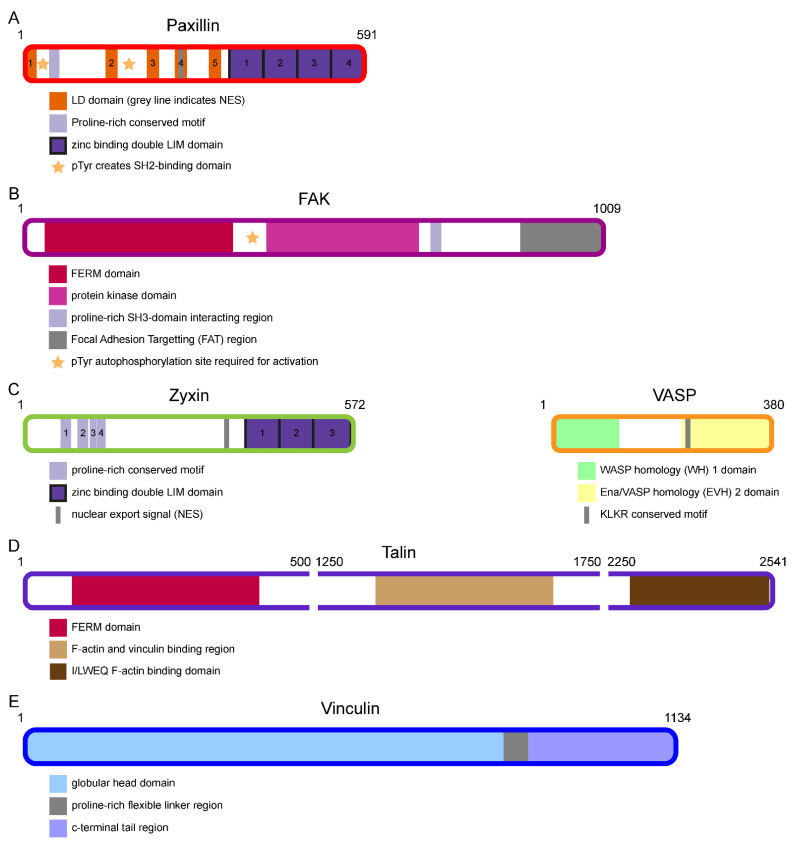
The structure of several intracellular FA proteins. (**A**–**E**) Schematic representation of the six intracellular FA proteins highlighted in this review, drawn to scale with respect to the amino acid backbone. Numbers indicate amino acid number along the protein backbone. Note that for talin (**E**), the visualised protein backbone is interrupted, and regions lacking relevant domains or structures (amino acid 500–1250 and 1750–2250) are not shown.

**Figure 4 biology-10-01189-f004:**
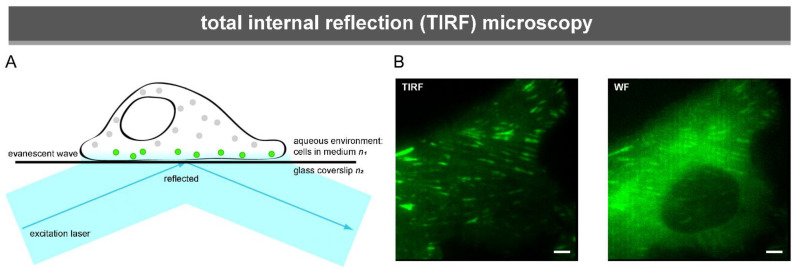
Total internal reflection microscopy. (**A**) Schematic overview of the light path during total internal reflection (TIRF) microscopy. The excitation laser exits the objective at such an angle that it is completely reflected at the glass–fluid interphase, formed by the glass coverslip and the aqueous environment of the cells in media, back into the objective. The reflecting laser light produces an electromagnetic field known as the evanescent wave, which exponentially declines. It is only powerful enough to excite fluorophores (green circles) in a thin layer ~100–200 nm upwards from the coverslip, leaving fluorophores at higher cellular locations unexcited (grey circles). (**B**) The same field of view, imaged in widefield (WF) or in TIRF mode, of a U2OS cell stably expressing paxillin-GFP. Scalebar 2 µm.

**Figure 5 biology-10-01189-f005:**
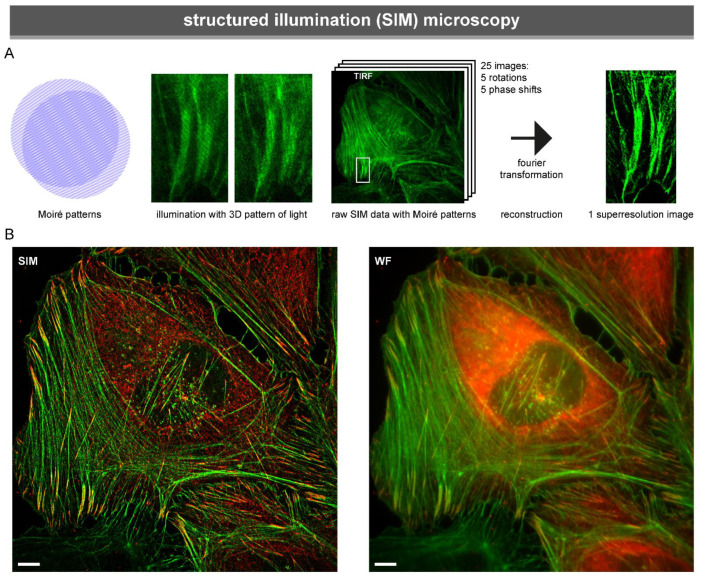
Structured illumination microscopy. (**A**) Schematic illustration of structured illumination microscopy (SIM). SIM makes use of Moiré patterns, an interference effect that occurs when two dense patterns are overlaid. Note that the formed Moiré pattern is always larger than the overlaid patterns. To create the Moiré patterns in SIM samples, they are illuminated with a three-dimensional pattern of light, which means parts of the sample are not illuminated (black stripes in the raw data). The pattern of light is rotated to allow for the reliable reconstruction (mainly through Fourier transformations) of a super-resolution image based on the collected Moiré patterns (orientation change of the black stripes in the raw data). The illumination pattern is also shifted to allow data collection from the whole image. In total, 25 raw images are typically collected to create 1 super-resolution image. (**B**) Reconstruction of the widefield (WF) or super resolution (SIM) image of a U2OS cell stably expressing paxillin-mCherry (red) and stained with phalloidin-CF405 (pseudocolour green). Scalebar: 5 µm.

**Figure 6 biology-10-01189-f006:**
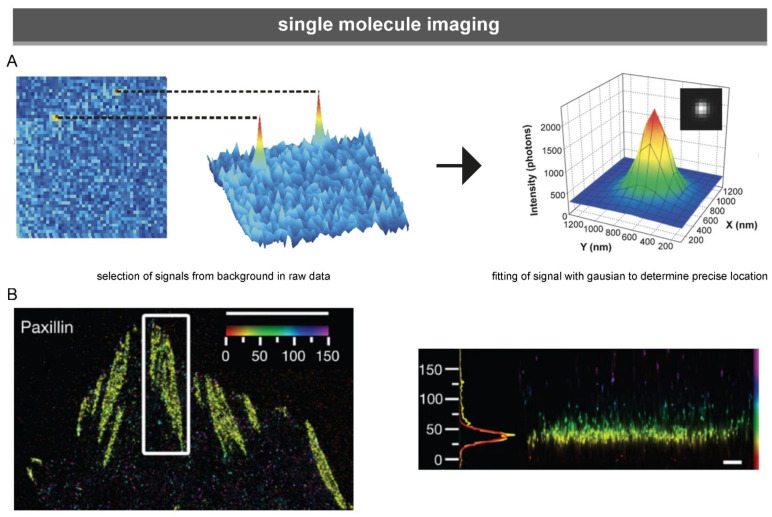
Single-molecule super-resolution microscopy. (**A**) Schematic illustration of single-molecule super-resolution microscopy. Adapted from van Royen et al. 2014 [156] (**B**) Single-molecule super-resolution image of paxillin-tdEos expressed in U2OS cells in top-view (**left**) and side view (**right**) of boxed area. Colour indicates vertical (z) position relative to the substrate (0) in nm. Side view includes corresponding z histogram and fit. Scale bar: 5 µm. Adapted from Kanchanawong et al. 2010 [37].

**Figure 7 biology-10-01189-f007:**
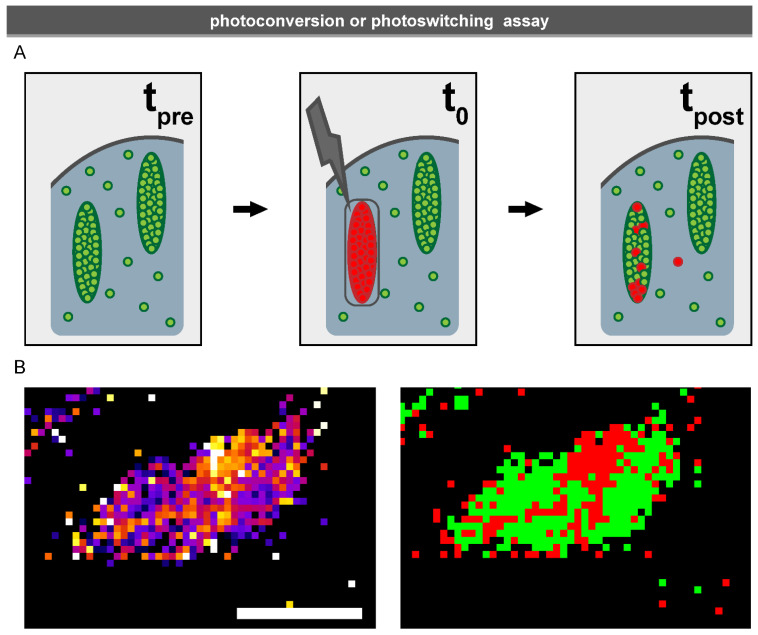
Photoconversion or photoactivation assay. (**A**) Schematic overview of a photoconversion or photoswitching assay to specifically visualise the proteins that are stably associated with a complex. The protein of interest tagged with a photoconvertible or photoswitchable fluorescent protein is expressed in the cell. Before photoconversion (t_pre_), the entire complex (e.g., an FA) is one colour (here, green). A small region tightly enclosing the protein complex is briefly exposed to a low intensity of 405 nm laser light (t_0_), converting the fluorescent proteins in this area to a different colour (here, red). After sufficient time has passed for the dynamic fraction to exchange, another image is taken (t_post_). As the volume exposed to the 405 nm laser is small, most fluorescent proteins remain unconverted (green); therefore, exchanging proteins from the converted complex will almost certainly be replaced by unconverted fluorescent proteins. In this way, at t_post,_ any remaining converted (red) fluorescence signal represents the stably bound fraction, while the unconverted signal represents the dynamically exchanging fraction. (**B**) Data from a photoconversion experiment where paxillin-mMaple3 is expressed in U2OS cells. Left image shows the ratio of converted signal at t_post_ over the converted signal at t_0_ on a pixel-by-pixel basis. Blue/purple indicates a low ratio representing the dynamically bound fraction, white/yellow a high ratio and the stably bound fraction. The right image is threshold showing the stably bound fraction in red and the dynamically bound fraction in green. Scalebar: 1 µm. Adapted from Legerstee et al. 2019 [58].

**Figure 8 biology-10-01189-f008:**
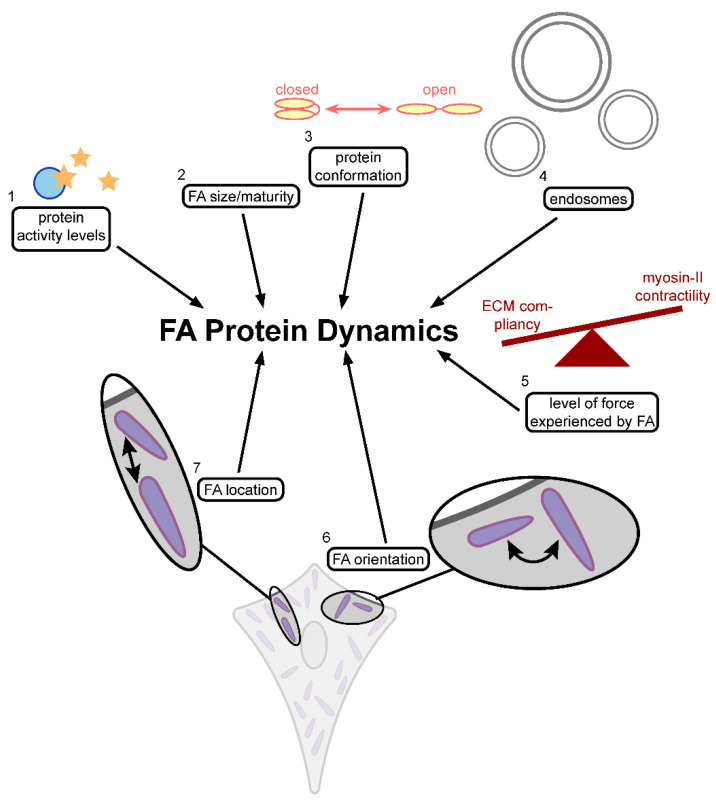
Factors reported to impact focal adhesion protein-binding dynamics. Schematic illustration of the diverse factors reported to correlate with focal adhesion protein-binding dynamics.

**Figure 9 biology-10-01189-f009:**
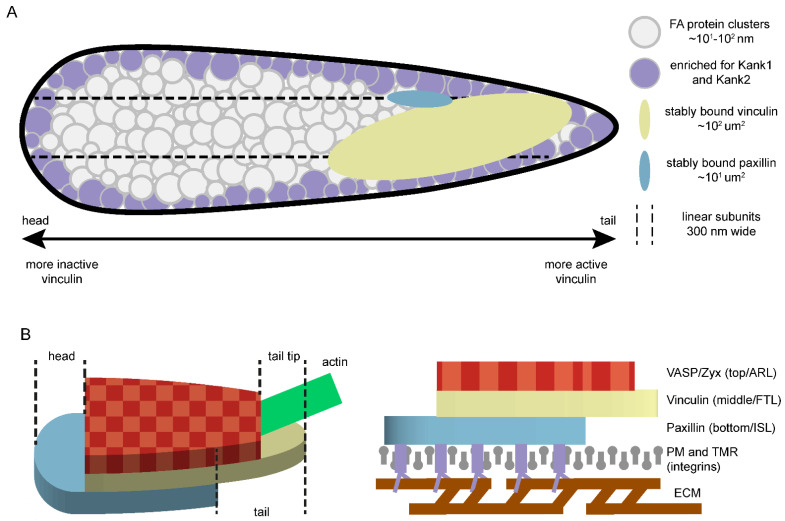
The nanoscale architecture of focal adhesions. (**A**) Schematic model of the nanoscale organisation of proteins within focal adhesions as seen from above. Head indicates the FA end closest to the edge of the ventral (adherent) portion of the plasma membrane. (**B**) Schematic side-view model of the nanoscale protein organisation within focal adhesions. PM is plasma membrane and TMR is transmembrane receptors.

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
