# Peer review of "A Layered View on Focal Adhesions"

_biology, 2021, doi:10.3390/biology10111189_

Round 1

Reviewer 1 Report

Well illustrated analysis of cytoskeleton components and well conducted review of the existing literature. I would like to see an extra section dedicated to clinical implications of this knowledge.

Author Response

We thank the reviewer for the kind comment and suggestion. We certainly agree that discussing clinical implications is an important aspect of an extensive review. However, to avoid repetition, we did not add a separate section on the topic, as it is difficult to discuss clinical implications without describing proper function. Dedicating a separate section to clinical implications requires to describe proper function again.

But as we certainly agree with the reviewer that our manuscript would benefit from more emphasis on clinical aspects we have expanded the last paragraph of subsection 5.3 (page 22) with more information on the clinical consequences of the rearrangement of focal adhesion layers in response to expression of the bacterial TarP-protein. We hope this satisfactorily addresses this issue for the reviewer.

Note that many more clinical implications were already discussed throughout the original manuscript:

A subsection devoted to the description of the general importance of focal adhesions to biological and clinical processes (subsection 1.2 “focal adhesions in health and disease”, p. 2) directly follows our brief introduction to focal adhesions.

For example, after we describe the functional domains and regions of the proteins highlighted in this review in subsection 2.1 to 2.3, we follow up with subsection 2.4 “the importance of individual FA proteins in vivo” (p. 7), describing the importance of these proteins to clinical processes. This section includes information gained from numerous knockout studies, as well as a detailed discussion of the particular importance of zyxin to the process of EMT in cancer.

Similarly, after we detail the correlation between FA orientation and location and its protein dynamics in subsection 4.6 “FA orientation and location and its protein dynamics”, we provide information on clinical consequences of FA location. (“Interestingly, another study points towards the biological consequences of disruption of the ratio of FAs far compared to close to the adherent membrane edge, as this ratio was increased in fibroblasts from individuals with bi-allelic pathogenetic variants in an enzyme important to the synthesis of heparan sulfate proteoglycans, important signalling molecules associated with the plasma membrane or found in the ECM.”, last paragraph section 4.6, p. 18).

Finally, after we discuss the current information on the layered nature of focal adhesions in subsection 5.3 ‘FA proteins are not strictly separated into layers’ we describe the biological and also clinical consequences of this knowledge in the last paragraph of this subsection (p. 22).

Reviewer 2 Report

The review from Legerstee and Houtsmuller is a nicely presented work examining molecular architecture of focal adhesion with an emphasis on the multilayer model. The work is well written and some original approach are present: 1) the selection of some well studied protein as models, 2) the inclusion of technical updates, 3) a critical evaluation of model weakness. English is correct although some phrasing let me perplex (but I’m not qualified enough for judging this).

The work could be improved:

  • Its mandatory to summarize at the beginning the concept of FA life cycle (assembly, maturation and dismantling) as this concept is diffusely cited but maybe unclear to non specialized readers.
  • Include a scheme of your 6 model proteins with domains and crucial interactors, this would help readers to follow the text
  • The techniques paragraph is interrupting the work. Could it be summarized and included in one or more side boxes like the one present in nature review articles? In this way the reader could chose if and when read them.
  • The chapters on protein dynamics are detailed but uneasy to follow and not summarized by figure 7. Maybe refining would help!

Author Response

The work could be improved:

  • Its mandatory to summarize at the beginning the concept of FA life cycle (assembly, maturation and dismantling) as this concept is diffusely cited but maybe unclear to non specialized readers.

We thank the reviewer for this comment. We already detailed the assembly and maturation processes of the FA life cycle in the first section of the original manuscript (last paragraph of section 1.3, p 3), and in Figure 2. In the revised manuscript we have added information on the disassembly of FAs, which we indeed previously omitted.

  • Include a scheme of your 6 model proteins with domains and crucial interactors, this would help readers to follow the text

We thank the reviewer for this excellent suggestion and have included this as novel Figure 3.

  • The techniques paragraph is interrupting the work. Could it be summarized and included in one or more side boxes like the one present in nature review articles? In this way the reader could chose if and when read them.

We are willing to do so, and indeed considered this during writing the manuscript, but were not sure about the required format. For this reason we have for now left it as it was.

  • The chapters on protein dynamics are detailed but uneasy to follow and not summarized by figure 7. Maybe refining would help!

We thank the referee for this comment and recognize the problem. To clarify the relationship between Figure 7 and the text, we have now refined our revised manuscript by numbering the different items in Figure 7 (now Figure 8) and referring specifically to these numbers in the text. We feel that in this way the reader is guided more smoothly through this section.

Reviewer 3 Report

This review by Drs. Karin Legerstee and Adriaan B. Houtsmuller on the organization of focal adhesions into layers is very well prepared, updated, and extensively addresses a relatively new topic, which is the layered organization of focal adhesions. This type of organization could only be detected thanks to advances in fluorescence microscopy techniques. As there are more than 200 components involved in focal adhesions, the authors restricted their review to the two most significant members of each layer. Thus, they provided an in-depth and detailed review of focal adhesions, their dynamics, as well as some insights into their regulation, as this is less well known. I highly recommend your publication. I noticed only a few typographical errors, which I highlight below.

Abstract line 11: “The last decade evidence…” shouldn't it be “In the last decade…”?

Line 120: I suggest re-word to “add significantly complexity to FA “

Line 129: There is an unnecessary comma in “…mechanistic, studies …”

Line 158: Is it possible to include vinculin in Fig 1, since it connects the three layers? It would be interesting.

Line 391: There is an unnecessary comma in “…parameters, on the mobility…”

Line 400: Please define kon’s and koff’s

Line 420: There is an unnecessary comma in “Another, much ...”

Line 477: There is another unnecessary comma.

Line 540: unnecessary comma.

Line 685: unnecessary dot in the legend to figure 8

Line 688: Please define HGF.

Author Response

We would like to thank the reviewer for the very kind comments, for reading our manuscript so thoroughly and for highlighting typographical errors. We rephrased all sentences indicated by the reviewer and removed all unnecessary commas and the unnecessary full stop in the legend. We apologize for omitting a definition of the terms kon and koff and for not defining the abbreviation HGF in our original manuscript and now included these definitions in our revised manuscript.

Abstract line 11: “The last decade evidence…” shouldn't it be “In the last decade…”?

This is indeed so. We adapted the text.

Line 120: I suggest re-word to “add significantly complexity to FA “ we reword

We adapted the text.

Line 129: There is an unnecessary comma in “…mechanistic, studies …”

We removed the comma.

Line 158: Is it possible to include vinculin in Fig 1, since it connects the three layers? It would be interesting.

The Figure shows a confocal image of actin and paxillin. We could add an image of vinculin but its distribution is not different from paxillin at the resolution of a confocal microscope. In the legend we added the statement: “note that the other FA proteins discussed here show identical distributions at confocal resolution”.

Line 391: There is an unnecessary comma in “…parameters, on the mobility…”

We removed the comma.

Line 400: Please define kon’s and koff’s

Indeed we should have better defined these parameters, which we now do: “…the sizes of these fractions and the associated on-rate (kon) - and off-rate (koff) (the rates at which the proteins bind to and dissociate from the focal adhesion complex),…”

Line 420: There is an unnecessary comma in “Another, much ...”

We removed the comma.

Line 477: There is another unnecessary comma.

We removed the comma.

Line 540: unnecessary comma.

We removed the comma.

Line 685: unnecessary dot in the legend to figure 8

We removed the dot.

Line 688: Please define HGF.

We apologize for this omission, which was due to final revisions of the original manuscript before submission. We have now defined HGF (Hepatocyte Growth Factor)